# Exposure to Forest Air Monoterpenes with Pulmonary Function Tests in Adolescents with Asthma: A Cohort Study

Davide Donelli [1,2], Michele Antonelli [3], Rita Baraldi [4], Anna Corli [4], Franco Finelli [5], Federica Gardini [6], Giovanni Margheritini [7], Francesco Meneguzzo [7,8,*], Luisa Neri [4], Davide Lazzeroni [9], Diego Ardissino [1,2], Giorgio Piacentini [6,10], Federica Zabini [8,*] and Annalisa Cogo [6,11]

1 Department of Medicine and Surgery, University of Parma, I-43121 Parma, Italy; davide.donelli@unipr.it (D.D.); diego.ardissino@unipr.it (D.A.)
2 Division of Cardiology, Azienda Ospedaliero-Universitaria di Parma, I-43126 Parma, Italy
3 Department of Public Health, AUSL-IRCCS of Reggio Emilia, I-42122 Reggio Emilia, Italy; michele.antonelli@ausl.re.it
4 Institute of Bioeconomy, National Research Council, 101 Via Gobetti, I-40129 Bologna, Italy; rita.baraldi@cnr.it (R.B.); anna.corli@unipv.it (A.C.); luisa.neri@ibe.cnr.it (L.N.)
5 Central Medical Commission, Italian Alpine Club, 19 Via E. Petrella, I-20124 Milano, Italy; franco.finelli55@libero.it
6 Institute Pio XII, 4 Via Monte Piana, I-32041 Misurina, Italy; direzionescientifica@misurinasma.it (F.G.); giorgio.piacentini@univr.it (G.P.); annalisa.cogo@unife.it (A.C.)
7 Central Scientific Committee, Italian Alpine Club, 19 Via E. Petrella, I-20124 Milano, Italy; giomarghe@yahoo.com
8 Institute of Bioeconomy, National Research Council, 10 Via Madonna del Piano, I-50019 Florence, Italy
9 Prevention and Rehabilitation Unit, IRCCS Fondazione Don Gnocchi, 3 Piazzale dei Servi, I-43121 Parma, Italy; davide.lazzeroni@gmail.com
10 Department of Surgical Sciences, Dentistry, Gynecology and Pediatrics, Pediatric Division, University of Verona, 10 Piazzale Ludovico Antonio Scuro, I-37134 Verona, Italy
11 Center for Exercise and Sport Science, University of Ferrara, 9 Via Savonarola, I-44121 Ferrara, Italy
* Correspondence: francesco.meneguzzo@cnr.it (F.M.); federica.zabini@cnr.it (F.Z.); Tel.: +39-392-985-0002 (F.M.); +39-333-379-2947 (F.Z.)

**Abstract:** Increasing evidence supports the direct healing effects of forests, partly attributed to the exposure to plant-emitted monoterpenes available in the forest atmosphere. The potential benefits on respiratory functions from inhaling monoterpenes have gained attention, especially due to the global rise in respiratory diseases. This study involved 42 asthmatic adolescents attending a summer rehabilitation camp at an Altitude Pediatric Asthma Center within a densely forested area in the Eastern Italian Alps. Volatile organic compound measurements indicated a pristine atmosphere, enabling the modeling of continuous hourly monoterpene concentration. The monoterpene concentration exposure and total inhaled dose were assessed over a 14-day stay, during which spirometry, lung oscillometry, and fractional exhaled nitric oxide were measured. Statistically significant correlations were observed between modifications in lung function parameters among asthmatic adolescents and monoterpene exposure. These findings suggest a potential localized airway effect that is specific to monoterpenes. This pilot cohort study might pave the way for further investigations into the therapeutic effects of forest monoterpenes on lung function tests, asthma, and the broader healing potential of forest environments.

**Keywords:** asthma; forest atmosphere; monoterpenes; respiratory functions; volatile organic compounds

## 1. Introduction

Exposure to forest environments has been associated with distinct and significant benefits for human health and well-being, encompassing both psychological and physiological aspects [1]. Among the short-term effects that are linked to forest exposure, evidence

has been gathered regarding immunological parameters [2] and inflammation-related factors [3,4]. Additionally, meta-analyses have reported positive impacts of interventions based in forest settings on the cardiovascular system [5].

Among various contributing factors, the effects of monoterpenes (MTs), emitted by plants and forest soil and available in the forest atmosphere, have been attributed a noteworthy role in enhancing psycho-physiological health [6–9]. Presently, our understanding of the health properties of MTs largely derives from in vitro studies, animal models, or indoor/laboratory research, often with limited sample sizes and concentrations that are much higher than those typically found in the forest atmosphere [10]. Recently, a few studies have explored the significant short-term psycho-physiological effects of individual exposure to plant-emitted MTs, and, in particular, a recent study identified a specific, significant, and dose-dependent effect of exposure to MTs in actual forest environments on the reduction of anxiety symptoms, based on a representative cohort of hundreds of participants in standardized forest therapy sessions [11]. In the latter study, it was suggested, in line with widely accepted evidence, that the improvement in anxiety symptoms might be associated with enhanced cardiovascular health. These findings confirmed preclinical evidence from animal studies, which indicated an anxiolytic activity associated with the monoterpene $\alpha$-pinene [12–14].

The upper respiratory tract serves as the entry point and main pathway for inhaled MTs during immersion in the forest atmosphere. It has been hypothesized that MTs enter the bloodstream through the nose and lung mucosa, affecting olfactory receptors and crossing the blood–brain barrier [15]. However, studies examining the pharmacokinetics and detailed mechanisms of inhaled MTs' action are scarce [16].

The potential benefits of MTs for respiratory functions have gained increased attention, especially considering the widespread prevalence of respiratory diseases [17]. Among these diseases, asthma affects more than 300 million individuals worldwide [18], and remains the most common chronic disease in childhood, showing an increasing trend in recent decades. Asthma is a complex condition resulting from the interplay of genetic, environmental, and lifestyle factors that contribute to its development and severity. Exposure to air pollutants ($CO$, $NO_2$, $O_3$, $SO_2$, ultrafine particulate matter, and volatile organic pollutants) can exacerbate respiratory health issues [19–21]. Exposure to outdoor air pollution was shown to potentially worsen pre-existing asthma, and numerous studies have indicated its role in the onset of asthma exacerbations [22].

The anti-inflammatory activity of a few MTs has been established [4,23,24], and the positive effects of certain MTs on asthma symptoms have been demonstrated in both animal models and preclinical studies [25–27]. Oral administration of eucalyptol (1,8-cineole) for 6 months has been associated with significant improvements in lung function, asthma symptoms, and quality of life in asthmatic patients, as compared to control groups [28].

The beneficial effects of staying in mountainous areas on asthma symptoms are well documented, and alpine altitude climate treatment (AACT) has been utilized for over a century for the treatment of asthma [29]. A recent study reaffirmed the positive impact of staying in a mountain environment on respiratory functions in asthmatic children, assessed through spirometry and oscillometer tests [30]. While the underlying mechanisms remain partially understood, the air characteristics that are related to mountain altitude (such as air density, temperature, and relative humidity), as well as a reduced exposure to allergens and air pollutants, have been proposed as key factors contributing to the benefits that are associated with the altitude environment. Factors that are related to the forest environment, directly linked to forest coverage, may also play a role in positive outcomes for asthma patients. Nevertheless, conclusive associations between improvements in pulmonary function and asthma symptoms, and specific features of the mountain and forest environments, have yet to be thoroughly investigated.

The objective of this pilot cohort study is to investigate whether the inhalation of the MTs that are available in the atmosphere of a mountain forest can positively impact lung function in asthmatic adolescents during a stay in an altitude rehabilitation center.

## 2. Materials and Methods

### 2.1. Study Area

The experimental site was the Pio XII Institute, an Altitude Pediatric Asthma Center located in northeastern Italy in a densely forested area in the Eastern Italian Alps, at about 1800 m a.s.l., around the natural alpine Lake Misurina (46°34′41″ N; 12°15′08″ E).

Lake Misurina, located in a protected area, precisely a Site of European Community Importance (SIC), denoted as SIC IT3230019, lies in a very scenic valley, surrounded by dolomitic peaks exceeding 3000 m a.s.l. Fed by streams, the lake is surrounded by the arboreal vegetation of the alpine forest, which, in the eastern area, reaches its banks. The dominant species are spruce (*Picea abies* (L.) Karst.), larch (*Larix decidua* Mill.), and stone pine (*Pinus cembra* L.), with scattered specimens of silver fir (*Abies alba* Mill.) and a few specimens of mountain pines (*Pinus mugus* Turra).

The main forest area is located to the east of the lake and extends in height up to more than 2000 m a.s.l., while it continues seamlessly down the valley for a few tens of km, often changing composition and merging with the biogenetic nature reserve of Somadida, and further down in large areas of cultivated forest. Locally, it is a relatively young forest, with the maximum diameter, measured by the authors across 50 trees belonging to the three dominant species, not exceeding 55 cm. The local forest stand shows a random arrangement of trees, suggesting a natural evolution.

Figure 1 shows the geographical location of the study area, a close-up view, the landscape with the Pio XII Institute in the foreground and the coniferous forest in the background, as well as the location of the air quantity measurement site inside the forest.

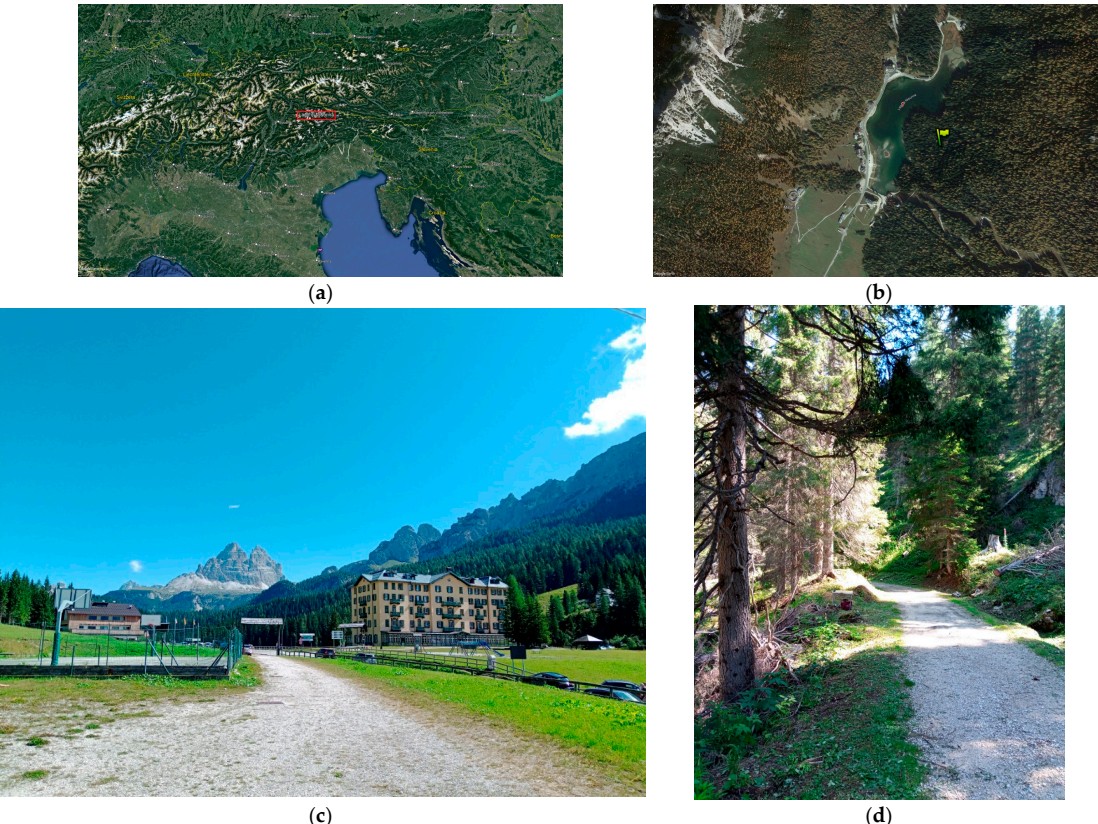

(**a**)　(**b**)

(**c**)　(**d**)

**Figure 1.** Geographical location and forest landscapes around Lake Misurina: (**a**) geographical location of Lake Misurina (rectangle with red border); (**b**) close-up view, with the flag indicating the site of atmospheric measurements, and the Pio XII Institute visible as the large building on the southern shore of the lake; (**c**) landscape with the Pio XII Institute in the foreground and the coniferous forest in the background; (**d**) location of the atmospheric parameter measurement site in the forest. Photos: F. Meneguzzo.

## 2.2. Design of the Study, Cohort, and Respiratory Measurements

A cohort of 42 asthmatic adolescents (mean age 15 $\pm$ 2 years, 12 females) who were admitted to the Pio XII Institute to participate in an intensive rehabilitation camp were included in this research. The patients were consecutively recruited in the period from July to September 2022, and were referred to the Institute by their pediatricians. All subjects were born and lived in low-altitude urban areas. All subjects had allergies: 24 were allergic to dust mites, 14 were allergic to both dust mites and pollen, and 4 were allergic to pollen. With regard to the therapy, 5 were administered only therapy as needed, 26 were taking medium-dose inhaled corticosteroids (ICS) plus long-acting beta2-agonists (LABA), 8 were taking low-dose ICS + LABA, and 3 were taking high-dose ICS + LABA. The basic information on the patients' characteristics is reported in Table 1.

**Table 1.** Basic parameters characterizing the cohort of patients considered for this study (BMI = body mass index; IBW = ideal body weight. In pediatrics, the Z-score, also known as the standard deviation score, is a statistical measure used to assess a child's growth and development by comparing their anthropometric data to a reference population of the same age and gender, allowing healthcare professionals to gauge how a child's measurements deviate from the norm).

| Variable | Mean | SE Mean | StDev |
|----------|------|---------|-------|
| Age | 14.6 | 0.31 | 2.01 |
| Height | 165.9 | 1.70 | 11.02 |
| Weight | 59.9 | 2.13 | 13.83 |
| Z-score | 0.3 | 0.15 | 1.00 |
| BMI | 21.6 | 0.57 | 3.72 |
| IBW | 52.5 | 1.45 | 9.41 |

Respiratory functions for all patients were assessed upon admission to the institute (T0) and after a 14-day stay (T1). All evaluations were conducted at the same time of day to mitigate circadian effects, precisely between 8 am and 10 am. Pre-study therapy remained unchanged throughout the study period. Following medical protocols, spirometry, lung oscillometry, and fractional exhaled nitric oxide (FeNO) measurements were performed to monitor changes in respiratory function, encompassing both the central and peripheral airways, as well as airway inflammation. All tests adhered to international guidelines [31,32], and were conducted using Master-screen IOS/SentrySuite$^{®}$ (VyAire Medical, Chicago, IL, USA), and an Analyzer CLD 88 sp (Eco Medics, Duernten, Switzerland).

Regarding the spirometric values, the forced expiratory volume in 1 s (FEV1), the forced vital capacity (FVC), the maximum mid-expiratory flow (FEF 25%–75%, the mean flow between 25% and 75% of forced expiration), and the FEV1/FVC ratio were considered for analysis. For the data derived from impulse oscillometry (IOS), parameters including respiratory system resistance at 5 Hz (R5) and 20 Hz (R20), the difference between R5 and R20 (R5–R20), the reactance at 5 Hz (X5), the resonant frequency of reactance (Fres), and the area under the reactance curve between Fres and 5 Hz (AX) were considered. Fractional exhaled nitric oxide (FeNO) was utilized to gauge airway inflammation [33].

In spirometry, FEV1 measures the volume of air that can be forcibly exhaled in one second after taking a deep breath, offering an as-assessment of pulmonary function. FVC is the total volume of air that can be forcibly exhaled after taking the deepest breath possible, which helps to evaluate lung capacity. Maximum mid-expiratory flow (FEF 25%–75%) refers to the average flow rate during the middle half (from 25% to 75% of expiration) of a forced expiration maneuver, often used to detect abnormalities in small airways. The FEV1/FVC ratio is an index that indicates how quickly the lung can empty and is useful in diagnosing and monitoring obstructive lung diseases. In impulse oscillometry (IOS), respiratory system resistance at 5 Hz (R5) and 20 Hz (R20) are measures of the opposition to airflow in the respiratory system at different frequencies. The difference between R5 and R20 (R5–R20) helps to specifically identify issues in the smaller airways. The reactance at 5 Hz (X5) measures the elastic and inertial properties of the respiratory system at a

frequency of 5 Hz. The resonant frequency of reactance (Fres) identifies the frequency where reactance becomes zero and switches from being capacitive to inductive. The area under the reactance curve between the Fres and 5 Hz (AX) gives an integrated view of reactance abnormalities. Finally, FeNO is used to gauge airway inflammation, providing insight into the presence of eosinophilic airway inflammation, which is relevant for diseases like asthma.

The study received approval from the Ethics Committee of the Province of Treviso and Belluno (26 May 2022, n. 1174). Written informed consent was obtained from parents or legal representatives.

### 2.3. Measurement of Volatile Organic Compounds

The total and individual concentrations of the volatile organic compounds (VOCs) were measured in the forested area frequented by children for the ordinary outdoor activities, about halfway through the route covered by the young patients. Within about 1000 m around the VOCs measurement point, the forest trees are dominated by spruce, followed by larch, with a few specimens of stone pine.

Biogenic volatile organic compounds (BVOCs), in which MTs represented the biologically active compounds, and anthropogenic volatile organic compounds (AVOCs), including mono-aromatic hydrocarbons such as benzene, toluene, ethylbenzene, and xylene isomers (BTEX), were quantified via air samplings. Each sampling extended for one hour and was conducted in triplicate at various times of the day, resulting in a total of 74 hourly measurements spanning from early July to late September 2022. Specifically, using portable pumps (PocketPump, SKC Inc., Eighty Four, PA, USA), air was directed into adsorption traps at a consistent flow rate of 200 mL/min, with each measurement involving 12 L of air. These traps, composed of inert metal tubes (8 cm × 0.3 cm i.d.) filled with Tenax TA and Carbograph 1TD (350 mg; 35/60 and 40/60 mesh, respectively, Markes International, Ltd., Llantrisant, UK), were stored at $-20\,°C$ until analysis. BVOCs, extracted from the traps via thermal desorption (unity series 2, Markes International, Sacramento, CA, USA), were injected into a 60 m capillary column (HP-1, 0.25 mm I.D.) coated internally with a 0.25 μm film of polymethylsiloxane (J&W Scientific USA, Agilent Technologies, Palo Alto, CA, USA). A 7890A gas chromatograph facilitated the BVOC separation, while eluted compounds were detected with a 5975C mass spectrometer (GC–MS, Agilent Technologies, Wilmington, USA). The oven temperature was held at $40\,°C$ for 10 min before progressively increasing to $280\,°C$ in increments of $5\,°C/min$. The helium flow rate into the column was set at 1.0 mL/min. In full-scan (SCAN) mode, sample ionization occurred at 70 eV. The analyzed mass range spanned from 30 to 350 $m/z$.

For BVOC identification, the retention times and mass spectra were gathered and cross-referenced with the NIST 11 library, using Agilent MassHunter Qualitative Analysis software. Following identification, compounds were quantified through external standard calibration, utilizing a gas cylinder with predetermined concentrations of distinct BVOCs (isoprene and $\alpha$-pinene; manufacturer: Apel-Riemer Environmental Inc., Broomfield, CO, USA). Relative response factors were computed for all components, and interpolation techniques were applied for a subset of compounds. Compound concentrations at experimental sites were calculated in μg/m$^3$ as the mean and standard error of the three replicates [34,35]. A total of 30 VOCs were detected, encompassing BVOCs (including MTs) and AVOCs (including BTEX).

### 2.4. Reconstruction of a Continuous Series of Hourly MT Concentrations

Based on the 74 hourly measurements collected from early July to late September 2022, a continuous series of hourly average MT concentration data was reconstructed. Due to the dominance of coniferous trees, the MT emission factor from plants was attributed only to pool emission, neglecting the de novo synthesized MTs that are also light-dependent. Thus,

the MT emission was considered to be dependent only on the temperature [36], using the approach of Guenther et al. (1993) [37], as per Equation (1):

$$\text{MTER} = \text{MTER}_s \cdot \exp\left[\beta(T - T_s)\right], \tag{1}$$

where MTER is the MT emission rate at the leaf temperature T(K), $\text{MTER}_s$ is the MT emission rate at the standard leaf temperature $T_s$(K) of 303.15 K, and $\beta(K^{-1})$ is an empirical coefficient depending on specific MTs, defined at the level of 0.1 $K^{-1}$ for the emission of the total MTs [36,37]. Due to the limited period of measurements and the dominant evergreen coniferous plants, no seasonal emission factors were considered. In Equation (1), T was assumed as the air temperature [36].

Although no comprehensive models were available for the reconstruction of the MT concentration in the forest atmosphere, despite early attempts [38], previous results have shown a strong dependence of atmospheric MT concentration on the MTER and the time of day, with the latter in turn connected to the surface atmospheric stability, such as in [39,40]. Hourly series of atmospheric MT concentrations were reconstructed via the means of a simple model based on the MTER and the surface atmospheric stability, and in particular, performing separate linear regressions of observed concentrations against the MTER for each atmospheric surface stability class. The reason is that, once emitted, MTs diffuse horizontally and dilute vertically in different ways according to the surface atmospheric stability conditions.

The atmospheric surface stability was reconstructed based on the Pasquill–Turner atmospheric surface stability classes [41]. Such classification was performed based on hourly weather data, available from a public source [42]: in the daytime, hourly average solar direct radiation and a 10 m wind speed, and at nighttime, a 10 m wind speed.

*2.5. Individual Exposure to MTs*

The exposure to MTs for each patient was evaluated by considering the average daily concentrations of MTs (ADMTc) during outdoor activity hours. To determine the total inhaled dose (TID), parameters such as the tidal volume (Vt), which refers to the volume of gas inhaled or exhaled in a single breath, and the minute ventilation (Ve), representing the average volume of gas entering or exiting the lungs per minute, were estimated. Subsequently, the total volume of air breathed during outdoor activities over the course of 14 days was calculated. This value was then multiplied by the ADMTc to derive the TID. Various approaches were employed to estimate the Ve and Vt, utilizing factors like heart rate, respiratory rate, age, sex, height, FVC, FEV1, body mass index (BMI), and ideal body weight (IBW) [43–47].

*2.6. Statistical Analysis*

Statistical analysis was conducted using Microsoft® Office Excel® and R Statistical Software version 4.4.1 [48]. Generalized linear models (GLMs) were employed to assess the correlation between exposure to total monoterpenes as the dependent variable and lung function parameters as outcomes. These models were adjusted for sex, age, BMI, and therapy to account for potential confounding factors. Two exposure metrics were utilized to enhance sensitivity: the average daily monoterpenes concentration (ADMTc) and the total inhaled dose (TID) over the 14-day period, considering outdoor hours. To compute the total inhaled dose, the minute ventilation (Ve) was estimated using body weight, height, and age. Multiple models of minute ventilation estimation were applied, as explained above, to ensure robustness in the analysis.

To further assess the treatment effects and control for potential confounders, we applied propensity score matching, thereby approximating a randomized experimental design. We addressed missing data by using mean imputation and random sampling for continuous variables, limited to a maximum of four missing data points per variable. Categorical variables, which had no missing values, remained unchanged. This approach resulted in the generation of two imputed datasets to bolster the robustness of our analysis.

The propensity scores were calculated using a minimal set of predictors—specifically, treatment type and sex—to avoid overfitting and multicollinearity issues, given the limited sample size. Utilizing the mean TID as an exposure cutoff, we then applied full matching to establish comparable groups of treated (above the mean) and untreated (below the mean) subjects. Subsequent to matching, we employed GLMs to evaluate the treatment's impact on the respiratory function outcome measures listed above. In these models, covariates included the TID exposure, the type of treatment the patients were on, and the sex of the subjects. Each imputed dataset was analyzed separately using this approach.

## 3. Results

### 3.1. Characterization of the Forest Atmosphere

Figure 2 shows the distribution of measured BTEX based on the concentration classes in the forest atmosphere around Misurina Lake, compared with the same distribution based on data that were collected during the years 2021 and 2022 at 33 mountainous sites distributed throughout Italy. The concentrations of volatile pollutants at Misurina Lake fell into the lowest classes, with 90% of the levels being lower than 50 ng/m$^3$, while more than 65% of the concentration levels across the mountain sites in Italy fell into classes above 100 ng/m$^3$. Furthermore, the main BTEX detected were benzene and toluene, with ethylbenzene and xylene isomers present only in traces. Overall, the forest atmosphere around Misurina Lake can be considered to be particularly pristine, with regard to volatile pollutants of both vehicular (traffic) and industrial origin. These characteristics help to avoid a possible confounding factor due to the action of the pollutants in worsening or eliciting asthma symptoms, affecting the outcomes and measured parameters.

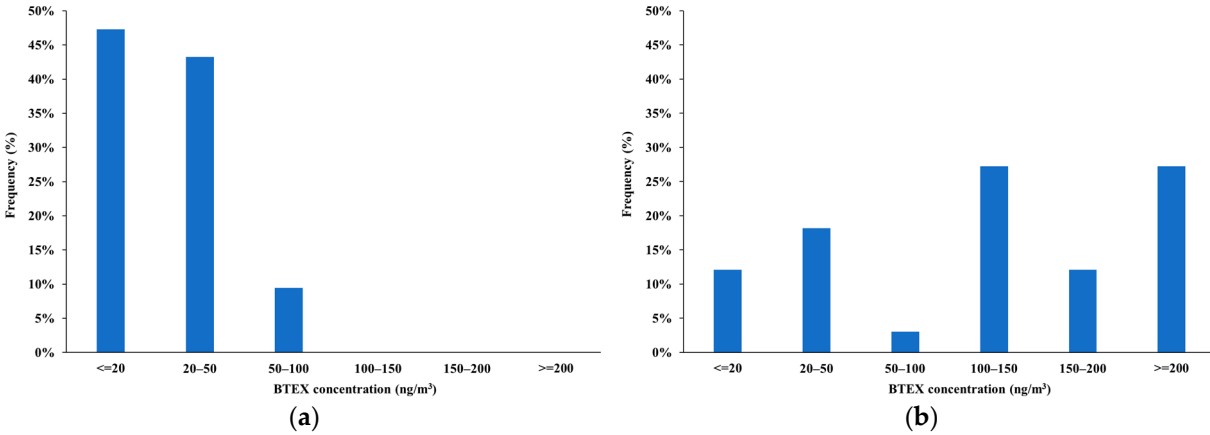

**Figure 2.** Distribution of BTEX based on atmospheric concentration classes: (**a**) at Misurina Lake; (**b**) at 33 mountain sites in Italy (2021–2022).

Figure 3 shows the distribution of MTs based on their concentration classes, compared with the same distribution based on data that were collected between 2021 and 2022 at 33 mountain sites in Italy, which was previously presented more synthetically [11]. The distribution of total MT concentrations at Misurina Lake was characterized by generally lower levels than at the other mountain sites. In particular, very few cases were observed with total MT concentration levels higher than 100 ng/m$^3$.

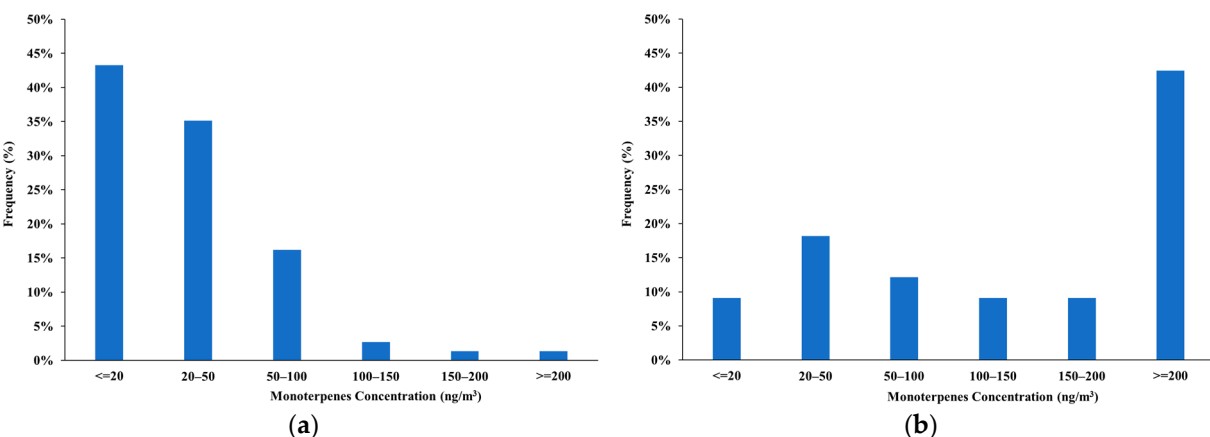

**Figure 3.** Distribution of MTs based on atmospheric concentration classes: (**a**) at Misurina Lake; (**b**) at 33 mountain sites in Italy (2021–2022).

Figure 4 shows the relative frequency of the measured concentrations of specific MTs with regard to the total MT concentrations. α-pinene accounts for about 53% of the total observed MT concentrations, as well as it explaining about 91% of the variance of the total observed MT concentrations (not shown), which agrees with the dominance of coniferous trees [40]. Sabinene, mostly emitted by beech trees [40], is the second MT by concentration, at about 16%.

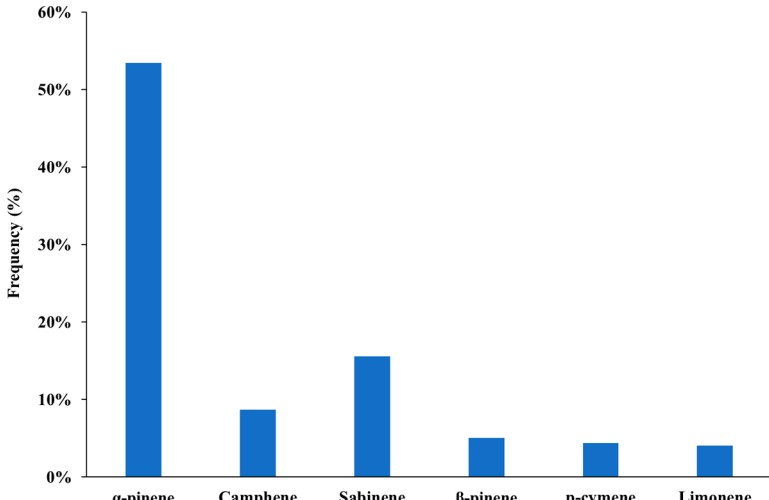

**Figure 4.** Relative frequency of the measured concentrations of specific MTs at Misurina Lake.

Figure 5 shows the hourly MT concentrations, computed according to the simple model described in Section 2.4, against the measured ones. The agreement was fairly good, despite a small pool of overestimated levels in the lower concentration range, corresponding to higher sabinene emissions, which behave differently from conifer-emitted MTs, in particular showing a strong dependence on photosynthetically active solar radiation [40], and a moderate underestimation in the higher concentration range. The explained variance was 80.3%.

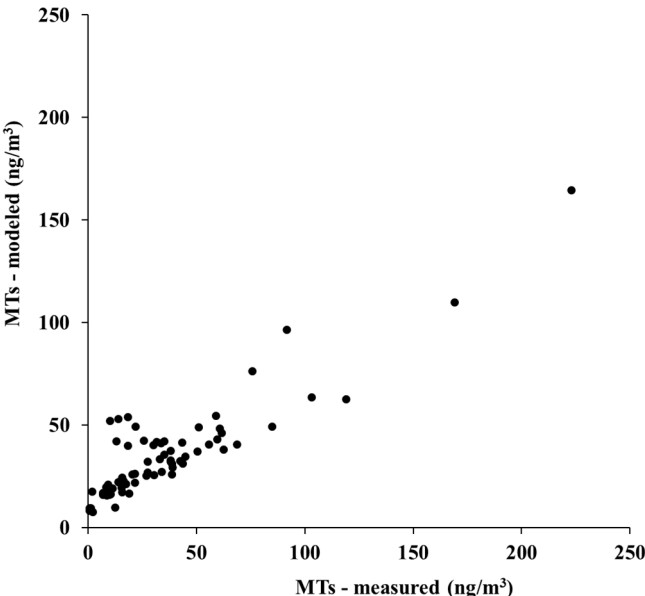

**Figure 5.** Modeled vs. measured MT concentrations.

Finally, Figure 6 shows the continuous series of model-reconstructed hourly MT concentrations since 1 July 2022. Apparently, the MT concentration was generally higher in July (up to hour 744) and lower in September (since hour 1489), which agrees with previous findings in a Mediterranean holm oak forest [49] and in a central European conifer forest [40], highlighting a remarkable variability.

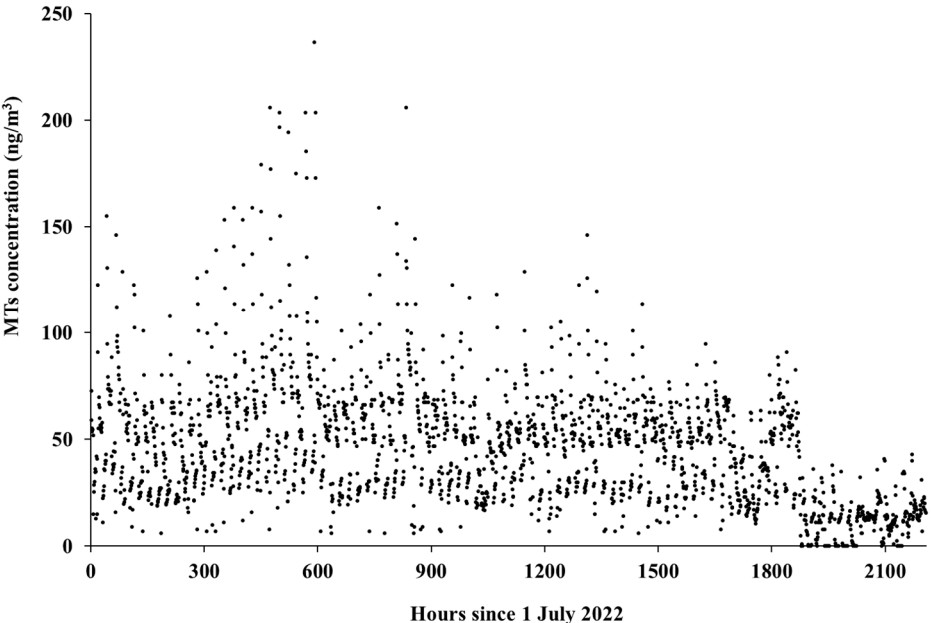

**Figure 6.** Continuous hourly series of model-reconstructed MT concentrations.

### 3.2. Association of MT Exposure with Pulmonary Functions

The average daily monoterpene concentration (ADMTc) ranged from 38 to 62 ng/m$^3$, while the total inhaled dose (TID) over 14 days ranged from 0.67 to 0.84 µg. Table 2 synthetizes the GLM results for both tested exposures (ADMTc and TID), showing pulmonary outcomes for which a statistically significant (or near-to-significant) association was found.

**Table 2.** Outcomes for which a statistically significant association of pulmonary functions with MT exposure was found. The meaning of IOS outcomes is as follows: R5, R20 = respiratory system resistance at 5 Hz and 20 Hz, respectively; R5–R20 = difference between R5 and R20; R5%, R20% = proportion of peripheral airway resistance to total airway resistance; X5 = reactance at 5 Hz; AX = area under the reactance curve between resonant frequency of reactance (Fres) and 5 Hz. Statistical significance is indicated with asterisks (*, $p < 0.05$; **, $p < 0.01$; ***, $p < 0.001$).

|  | Outcome | Estimate [a] | Std. Error | t | *p* Value |
|---|---|---|---|---|---|
| ADMTc exposure | FEV1/FVC% | 302 | 147 | 2.05 | 0.05 |
|  | FEF25/75% | 13 | 558 | 2.36 | 0.02 * |
|  | R5 | −5.76 | 2.21 | −2.61 | 0.01 ** |
|  | R5% | −1038 | 532 | −1.95 | 0.05 |
|  | AX | −28.6 | 11.7 | −2.45 | 0.02 * |
| TID exposure | FEV1/FVC% | 10.6 | 4.9 | 2.2 | 0.04 * |
|  | FEF25/75% | 46.6 | 19.6 | 2.4 | 0.02 * |
|  | R5 | −0.27 | 0.07 | −3.78 | <0.001 *** |
|  | R5% | −70 | 34 | −2.1 | 0.04 * |
|  | R20 | −0.14 | 0.06 | −2.32 | 0.03 * |
|  | R20% | −39 | 19 | −2.1 | 0.04 * |
|  | X5 | 0.07 | 0.03 | 2.4 | 0.02 * |
|  | AX | −1.38 | 0.4 | −3.65 | <0.001 *** |
|  | R5-R20 | −0.12 | 0.04 | −2.94 | 0.006 ** |

[a] The estimates refer to the change in outcome for a 1 µg/m$^3$ increase in ADMTc or a 1 µg increase in TID exposure.

The outcomes derived from TID exposure corroborate the findings obtained from ADMTc exposure. Additionally, associations with further pulmonary oscillometry parameters were identified. No significant associations were observed between the evaluated exposures and FeNO, FEV1, FVC, blood pressure, or peripheral saturation. Importantly, in the supplementary robustness tests, employing various minute ventilation estimation techniques, significant associations were observed with FEV1 at the 14-day mark. However, the more conservative model was retained.

Figure 7 shows the log estimates of the data that are reported in Table 2 for the ADMTc and TID exposures.

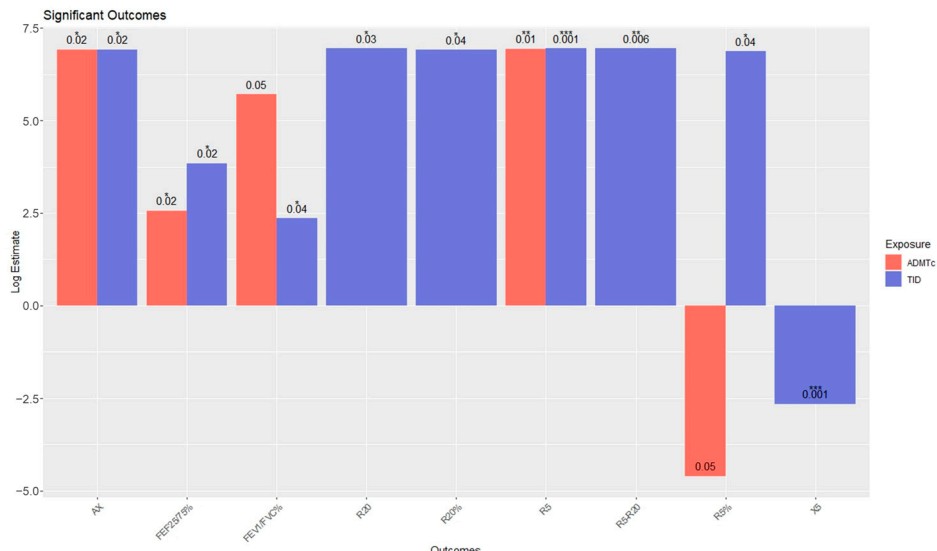

**Figure 7.** Log estimates for effects on pulmonary function parameters of ADMTc and TID exposures to MTs. Statistical significance is indicated with asterisks (*, $p < 0.05$; **, $p < 0.01$; ***, $p < 0.001$).

Table 3 summarizes the significant or nearly significant impacts of above-average TID exposure on multiple measures of pulmonary function, based on propensity-matched GLMs. Propensity matching yielded an excellent balance (standardized mean differences for the covariates < 0.01) among 18 treated patients vs. 24 control patients. Both imputation methods consistently demonstrated the effects of TID exposure on respiratory resistance (R5% and R20%), reactance at 5 Hz (X5%), and resonant frequency (Fres%). A nearly significant impact was also observed on the forced expiratory flow (FEF25/75%).

**Table 3.** Outcomes for which a statistically significant (or nearly significant) specific effect of MT TID exposure on pulmonary functions was found after propensity score matching. The first row of results refers to the model performed with missing values imputed with the mean, while the second row refers to the model performed with missing values imputed with the random sample. Statistically significant results are indicated with asterisks (*, $p < 0.05$; **, $p < 0.01$).

|  | Outcome | Estimate [a] | Std. Error | t | *p* Value |
|---|---|---|---|---|---|
| TID exposure | FEF25/75% | 11.4 | 6.2 | 1.8 | 0.07 |
|  |  | 12.7 | 7.0 | 1.8 | 0.07 |
|  | R5% | −26 | 8.6 | −3 | 0.004 ** |
|  |  | −24.2 | 8.9 | −2.7 | 0.01 ** |
|  | R20% | −15.8 | 7 | −2.3 | 0.03 * |
|  |  | −17.8 | 7.4 | −2.4 | 0.02 * |
|  | X5% | −47.7 | 22.1 | −2.2 | 0.03 * |
|  |  | −50.3 | 22.3 | −2.3 | 0.03 * |
|  | Fres% | −30.1 | 14.2 | −2.1 | 0.04 * |
|  |  | −27.5 | 14.6 | −1.9 | 0.04 * |

[a] The estimates refer to above-average TID exposure.

The propensity-matched analysis revealed that TID exposure has a favorable impact, predominantly on respiratory reactance, specifically at frequencies of 5 Hz and 20 Hz, as well as resonant frequency, rather than having a significant impact on spirometric outcomes. This points to a more localized beneficial effect of TID exposure on the airways. All these findings seem to suggest the presence of a specific effect of monoterpene inhalation on respiratory function, with beneficial effects for asthmatic patients. However, this specific effect appears to be more pronounced in terms of peripheral airway dilation, and possibly a synergistic effect in conjunction with the ongoing therapy.

## 4. Discussion

In this study, we demonstrated a statistically significant association between certain lung function parameters and total MT exposure in asthmatic adolescents during a 14-day stay at the Pio XII Institute in Misurina, situated near an alpine forest and a natural lake. Notably, we observed meaningful associations between the 14-day TID and the enhancement of impulse oscillometry (IOS) parameters related to small airway patency. These findings are in line with previous research, indicating improved lung function following periods spent in comparable high-altitude mountain regions [29,30,50]. To the best of our knowledge, only one study has previously shown IOS improvement [30]; other research typically focused solely on spirometric parameters, typically FEV1, FVC, and FEF25-75, for assessing lung function in asthmatic children in mountain environments.

Enhancements in airway resistance and reactance hold significant importance, as peripheral airway impairment is recognized to precede asthma exacerbations and predict their likelihood [51,52]. The positive clinical and functional changes observed in asthmatic participants during altitude stays have been attributed primarily to allergen avoidance, especially with regard to dust mites and pollen. At higher elevations, the pollen season is shorter, and Dermatophagoides struggle to thrive above 1500–1800 m a.s.l. due to the lower humidity and temperatures.

While many of the asthmatic participants had allergies to pollen and Dermatophagoides, clinical and functional improvements were noted in non-allergic individuals as well. This suggests that, aside from allergen reduction, other factors play a pivotal role in alleviating the inflammation that underlies asthma. A potential positive impact of reduced pollution has been proposed, considering the known connection between environmental pollution and lung function, particularly in asthmatic children [53].

Our data additionally reveal consistently low atmospheric BTEX concentrations throughout the study period, thereby sidestepping potential confounding factors due to the impact of pollutants on asthma symptoms. These insights partly elucidate the improvements, aligning with the non-pharmacological approach of asthma trigger avoidance, which involves irritants or inflammation exacerbators in the airways [54]. This study thus contributes novel insights by revealing a plausible underlying mechanism, tied to atmospheric MT concentrations and their role in favoring small airway dilation.

While this study offers a pioneering insight into MTs' distinct impact on asthmatic adolescents' lung functions, it bears limitations. Firstly, the relatively small sample size and the single-center nature are limiting factors; expanding this investigation to other altitude-based asthma rehabilitation centers could bolster the findings. Another limitation pertains to the accuracy of modeling a continuous hourly MT concentration series based on discrete measurements. Finally, the pharmacokinetics and pharmacodynamic healing mechanisms of MTs inhaled in forest atmospheres still remain to be clarified, both for short-term exposures, as seen in Donelli et al. 2023 [11], and for more extended durations, as observed in this study.

## 5. Conclusions

This pilot study has substantiated that exposure to plant-emitted MTs in a forest atmosphere significantly enhances lung function among asthmatic adolescents, potentially via localized airway effects. This therapeutic influence gains further credence from the more robust and substantial associations observed with the total dose of inhaled MTs compared to the simple exposure to their air concentration. Nonetheless, further research is warranted to validate these findings and unravel the mechanisms underpinning the therapeutic impact of forest MTs on asthma symptoms.

This study contributes additional evidence to the increasingly explored direct healing potential of forest environments. Additionally, the relatively modest average MT concentrations imply the efficacy of prolonged exposure to a forest atmosphere, expanding on the dose-dependent effect that has been noted in brief exposures to different MT concentrations in relation to anxiety symptoms [11]. Such insights may guide the planning of forest-immersion therapeutic interventions, even during less favorable seasons like spring, fall, or winter.

Within the context of forest medicine, this research, grounded in a forest-centric approach, seeks to deepen the comprehension of forest environments' therapeutic role, a domain that still remains to be explored. In addition to evaluating the health benefits arising from exposure to forest ecosystems, this study also concentrated on scrutinizing the environmental attributes and components associated with these health benefits. This twofold approach aims to advance the adoption of standardized guidelines for public health interventions and enhance forest management practices.

**Author Contributions:** Conceptualization, D.D., F.M., G.M. and A.C. (Annalisa Cogo); methodology, D.D., D.L. and F.M.; software, D.D.; validation, M.A., F.F., F.G., D.A. and G.P.; formal analysis, D.D.; investigation, R.B., A.C. (Anna Corli), F.G., G.M., L.N. and G.P.; resources, G.M., F.Z. and A.C. (Annalisa Cogo); data curation, F.M., R.B., L.N., A.C. (Anna Corli) and A.C. (Annalisa Cogo); writing—original draft preparation, D.D., F.M., M.A., R.B., L.N., F.Z. and A.C. (Annalisa Cogo); writing—review & editing, D.D., F.M., M.A., R.B., L.N. and D.L.; visualization, D.D. and F.M.; supervision, G.M., F.M., F.Z., D.A. and A.C. (Annalisa Cogo); project administration, G.M., F.Z. and A.C. (Annalisa Cogo). All authors have read and agreed to the published version of the manuscript.

**Funding:** This research received no external funding.

**Institutional Review Board Statement:** This study was designed and conducted in compliance with the Declaration of Helsinki and its subsequent modifications. The study received approval from the Ethics Committee of the Province of Treviso and Belluno (26 May 2022, n. 1174). All participants voluntarily provided their informed consent prior to enrollment and were free to withdraw from the study at any time. Written informed consent was obtained from parents or legal representatives; all the information was strictly handled according to the EU privacy regulation ("The General Data Protection Regulation"—GDPR 2016/679).

**Data Availability Statement:** All data are available by contacting the corresponding author.

**Acknowledgments:** Giovanni Indiati (CNR) and Francesco Centritto (CNR-IBE) are gratefully acknowledged for performing part of the field measurements.

**Conflicts of Interest:** The authors F. Gardini (Scientific Director at the Institute Pio XII, Misurina, Belluno, Italy), G. Piacentini (Scientific Consultant at the Institute Pio XII, Misurina, Belluno, Italy), and A. Cogo (Health Manager at the Institute Pio XII, Misurina, Belluno, Italy) declare a potential conflict of interest; however, Institute Pio XII had no involvement or role in the study design; on the collection, analysis, and interpretation of the data; in the writing of the report; and the decision to submit the report for publication. All the other authors declare no conflict of interest.

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
