# Peer review of "Exposure to Forest Air Monoterpenes with Pulmonary Function Tests in Adolescents with Asthma: A Cohort Study"

_forests, doi:10.3390/f14102012_

Round 1

Reviewer 1 Report

In this study, 42 asthmatic adolescents attended a summer rehabilitation camp at an Altitude Pediatric Asthma Center within a densely forested area in the Eastern Italian Alps. Monoterpene (MTs) concentration were measured and hourly concentration were simulated. The exposure and total inhaled dose of MTs were assessed over a 14-day stay. The correlations were also investigated between modifications in lung function parameters among asthmatic adolescents and MTs exposure. However, I would not recommend the publication of this paper.

 1)      The authors claimed that statistically significant correlations were observed between modifications in lung function parameters among asthmatic adolescents and MTs exposure. While correlations is not a quite convincing evidence.

2)      The sample size too small?  42 young attendants? Design of the study, cohort, and respiratory measurements needs to be further described.

3)      Other traditional pollutants, including PM2.5 and its components, SO2, NOx, and O3, were not well reported.

4)      The average daily concentrations of MTs varied a lot in different plant types. While in the current study, only limited sample were analyzed.

Reviewer 2 Report

This manuscript by Donelli et al. aims to investigate the short-term healing effects of forests through biogenic monoterpenes emissions. The topic addressed in this manuscript is highly relevant to scientists, doctors, and forest managers. However, in my opinion, this manuscript has several shortcomings and requires a major revision before it can be considered for publication in Forests.

Major criticisms:

1. My major concern regarding this study is the absence of a suitable control group in the cohort experiment. As mentioned by the authors themselves, the beneficial effects of staying in mountainous areas could be attributed to various factors, such as favorable meteorological conditions and reduced exposure to allergens and air pollutants. Without a control group, the conclusion drawn about the significant association between lung function parameters and exposure to MTs is not reliable.

 2. Page 4 L187-199: Please justify the assumption that all MTs are pool emissions. As demonstrated by many previous studies, a significant proportion of monoterpenes emissions are light-dependent. One possible approach is to validate your MT monitoring data against the simulated results of Genther model.

 Some specific comments and suggestions:

1. Page 3 L126: yo should be yr?

2. Figure 1: Please include a map in Figure 1 to illustrate the geographical location of the study area.

3. Page 4 L133: Please give the exact time when these evaluations were conducted.

Reviewer 3 Report

A reference is needed in line 309

I suggest to make change in Table 2. I will advise the authors to show  adj R2 of the model, beta and the significans with asterics. 

The terms R5, R5%, R20, R20%, X5, AX and R5R20 are all IOS parameters, it should be shown in table 2. The readers had to go back to the methods chapter to understand the table.  

I will strongly recommand that the authors showed one or two figures from line 294. The figures should show the correlation between MT and one or two of the health outcomes. It can be one of the lung function measures associated with TID exposure and one of the IOS parameters associated with TID exposure.  

Reviewer 4 Report

Thank you for the opportunity to review this article. The subject matter is important, as is having more research conducted using some of the volatile organic compounds measuring and monitoring techniques. The authors are accurate that: “Potential benefits of MTs for respiratory functions have gained increased attention, especially considering the widespread prevalence of respiratory diseases.”

The authors mention in the abstract and the conclusion that this is a pilot study. I think it needs to be mentioned in the introduction, too. I hope there is more follow up because it is a small sample size. At the same time, more research, especially expanding measurement techniques is needed in this area. Their trying and refining samplings of “biogenic volatile organic compounds (BVOCs), in which MTs represented the biologically active compounds, and anthropogenic volatile organic compounds (AVOCs), including mono-aromatic hydrocarbons such as benzene, toluene, ethylbenzene, and xylene isomers (BTEX), were quantified via air samplings” is excellent. Follow up research is recommended.

A strength of the article is naming the specific tree species in the area.  For studies like this, it is important to know the species because more detail is needed regarding specific tree species and volatile organic compounds emission—not all vegetation and trees’ emissions produce the same health results. [“Dominant species are spruce (Picea abies), larch (Larix decidua), and stone pine (Pinus cembra), with scattered specimens of silver fir (Abies alba), and a few specimens of mountain pines (Pinus mugus).”]

Table 1: Main characteristics of the patients—needs a more descriptive title.  It also

needs a table note to let the readers know what BMI and IBW are, and the like.  Please carry this comment through with your other tables and figures, too.

I appreciate the authors, naming specific limitations of this study their study. It is important to emphasize to the readers this is a pilot study with a small sample. The measurement techniques are still being refined. I think the authors did a good job of not overstated results, and I encourage them to be continually mindful of keeping their results in perspective.

Round 2

Reviewer 2 Report

The authors have made modifications and improvements according to the suggestions, and I have no other questions.

Reviewer 3 Report

This article is interesting and provide new knowledge in this research field.